# Scope, context and quality of telerehabilitation guidelines for physical disabilities: a scoping review

Krithika Anil ,[1] Jennifer A Freeman,[1] Sarah Buckingham,[1] Sara Demain,[1,2] Hilary Gunn,[1] Ray B Jones ,[3] Angela Logan,[4] Jonathan Marsden,[1] Diane Playford,[5] Kim Sein,[1] Bridie Kent[3,6]

¹School of Health Professions, Peninsula Allied Health Centre, University of Plymouth, Plymouth, UK
²School of Health Sciences, University of Southampton, Southampton, UK
³School of Nursing and Midwifery, Plymouth University, Plymouth, UK
⁴Stroke Rehabilitation, Royal Devon and Exeter NHS Foundation Trust, Exeter, UK
⁵Warwick Medical School, University of Warwick, Coventry, UK
⁶Innovations in Health and Social Care: A JBI Centre of Excellence, Plymouth University, Plymouth, UK

**Correspondence to**
Dr Krithika Anil;
krithika.anil@plymouth.ac.uk

## ABSTRACT

**Objective** To identify the available guidance and training to implement telerehabilitation movement assessments for people (adults and children) with a physical disability, including those recovering from COVID-19.

**Design** Rapid scoping review.

**Included sources and articles** PubMed, CINAHL, PsychInfo, Cochrane, Embase, Web of Science, PEDro, UK Health Forum, WHO, National Archives and NHS England were searched using the participant–concept–context framework from 2015 to August 2020. Primary studies that recruited individuals with physical disabilities and guidance documents aimed at providers to implement movement-related telerehabilitation were included.

**Results** 23 articles (11 primary research studies, 3 systematic reviews and 9 guidance documents) were included out of 7857 that were identified from the literature search. Two main issues were found: (1) telerehabilitation guidance (from both research studies and guidance documents) was not specific to movement-related assessment and (2) most primary research studies provided neither guidance nor training of movement-specific assessment to practitioners. Of the COVID-19 related guidance, two articles reported COVID-19 management that only referred to identifying COVID-19 status without references to specific movement-related guidance.

**Conclusions** Telerehabilitation guidance and training have existed pre-COVID-19, yet the lack of specific movement-related information and provider support is surprising. This gap must be addressed to optimise effective implementation of remote assessments for those with physical disabilities.

**Review registration** Open Science Framework: osf.io/vm6sp.

## INTRODUCTION

Since the outbreak of the COVID-19 pandemic, health services across the world have rapidly adapted to a new way of working. They have embraced the need to use a 'digital first' approach to maintain delivery of care as best as possible, while protecting both staff and patients.[1] In the field of rehabilitation, this approach is typically referred as 'telerehabilitation', defined as: the delivery

### Strengths and limitations of this study

► A wide variety of articles were reviewed due to extensive searches for grey literature and no language restrictions.
► Despite no language restrictions, included articles were biased towards the English language.
► Some articles within this review may have been updated since the initial search and others have likely emerged since.

of rehabilitation services via information and communication technologies.[2] This includes a broad range of services from assessment and monitoring to education and consultation. There is a general acknowledgement that, beyond the pandemic, remote delivery of services will be an integral part of everyday practice within future health and social care systems. Ensuring that the delivery of such services is both effective and equitable is pertinent, given the identified 'tsunami' of rehabilitation need[3 4] for people with a wide range of conditions from across the lifespan (including those recovering from COVID-190).[5] It is especially important to note that the practical application of telerehabilitation will not be the same across all conditions. More complicated conditions (eg, those with comorbidity) will likely require additional support than less complicated conditions. This additional consideration further demonstrates the need for comprehensive training and guidance for telerehabilitation.

Key to targeting an effective and personalised rehabilitation plan is the need for a comprehensive, detailed and valid assessment,[6] with ongoing monitoring and evaluation as needs change. When the primary concerns relate to physical disability, an assessment of aspects such as dynamic posture, balance and movement is fundamental. This therefore raises important questions related

to how these assessments can be undertake remotely, while ensuring they are both safe and effective. For instance, is it possible to adequately and safely assess the balance of an individual who is falling/at risk of falls? Or the source of shoulder pain? Or the relative impact of spasticity and weakness on movement? Do the additional practical challenges such as slow internet speeds and static and restricted camera angles mean that an assessment of dynamic movement is too restricted to inform management plans? Are the clinical skills required for this type of remote assessment different to those used for face-to-face assessment? And if so, is there specific guidance and training related to these aspects of remotely based physical assessments to assist practitioners with implementing this effectively and efficiently? It was our observation that questions such as these have been raised by rehabilitation organisations, professional networks and individual practitioners and require urgent answers, hence the need for this rapid scoping review.

This scoping review formed part of an overall research project aimed at developing and evaluating a training package to support the remote assessment and management of people with physical disability (UK RI-NIHR, MRC, COVID-19: MR/V021060/1). It was not appropriate to involve patients or the public in the design, or conduct, or reporting of this research.

## Review question and objectives

What guidance and training is available to implement telerehabilitation movement assessment for people (children and adults) with a physical disability, including those recovering from COVID-19? The review objectives were to identify:

► The contexts and types of telerehabilitation assessments delivered in a health and social care setting for people with a physical disability (including those recovering from COVID-19).
► Any specific guidance and/or training available for delivering telerehabilitation movement-related assessments.
► Specific recommendations regarding movement-related assessments for effective implementation of telerehabilitation guidance.

## METHODS
### Type of review

This scoping review used the Joanna Briggs Institute (JBI) approach.[7] This approach seeks to explore the breadth or extent of the literature, map and summarise the evidence and inform future research.[8] The Preferred Reporting Items for Systematic Reviews and Meta-Analyses (PRISMA) checklist for scoping reviews was used for the review (see online supplemental material 1 titled 'SM – PRISMA Statement for Scoping Reviews'). There is no registration database specific for scoping reviews, thus the review was uploaded to Open Science Framework: osf.io/vm6sp.p9.

## Patient and public involvement
No patients involved.

## Inclusion criteria
The inclusion criteria were developed using the participant–concept–context framework.[9]

### Participants
The review considered articles that included individuals of all ages with a physical disability. Physical disability was defined as any physical condition that affects a person's mobility, physical capacity, stamina or dexterity. The review included studies of participants with a physical disability who had any duration and severity of physical disability. There was no age restriction. Guidance and policy documents aimed at undertaking telerehabilitation with individuals with physical disabilities were also included.

### Concept
Studies and policies that examined or described guidance and training regarding implementation of telerehabilitation movement assessments were considered. *Guidance* included checklists, tools, outcome measurement batteries, protocols, guidelines, risk assessments, exemplars and appraisals. *Training* included interactions, communications and resources with the aim to improve and/or inform the process of delivering telerehabilitation. Guidance and training must have involved movement evaluation; that is, the evaluation of how people move around, perform and function. This included self-administered and practitioner administered telerehabilitation. Studies and policies that did not involve movement evaluation were excluded.

### Context
This review considered studies that provided guidance and training in health and social care contexts; that is, telerehabilitation must have been provided by a practitioner, system or institute.

### Types of literature
The following study designs were included: randomised controlled trials (RCTs), non-RCTs, quasiexperimental, prestudies and poststudies, case studies, observational studies, systematic reviews and qualitative studies. Grey literature regarding policies from recognised UK-based national and international institutes were also considered. Opinion pieces, non-systematic literature reviews and other grey literature were excluded. No language restrictions were implemented.

### Date restriction
Publications from 2015 to August 2020 were included in this review because technologies and related-guidance older than this period were considered out of date due to the rapid speed of technology development in areas such as robotics, artificial intelligence and telepresence.[10 11]

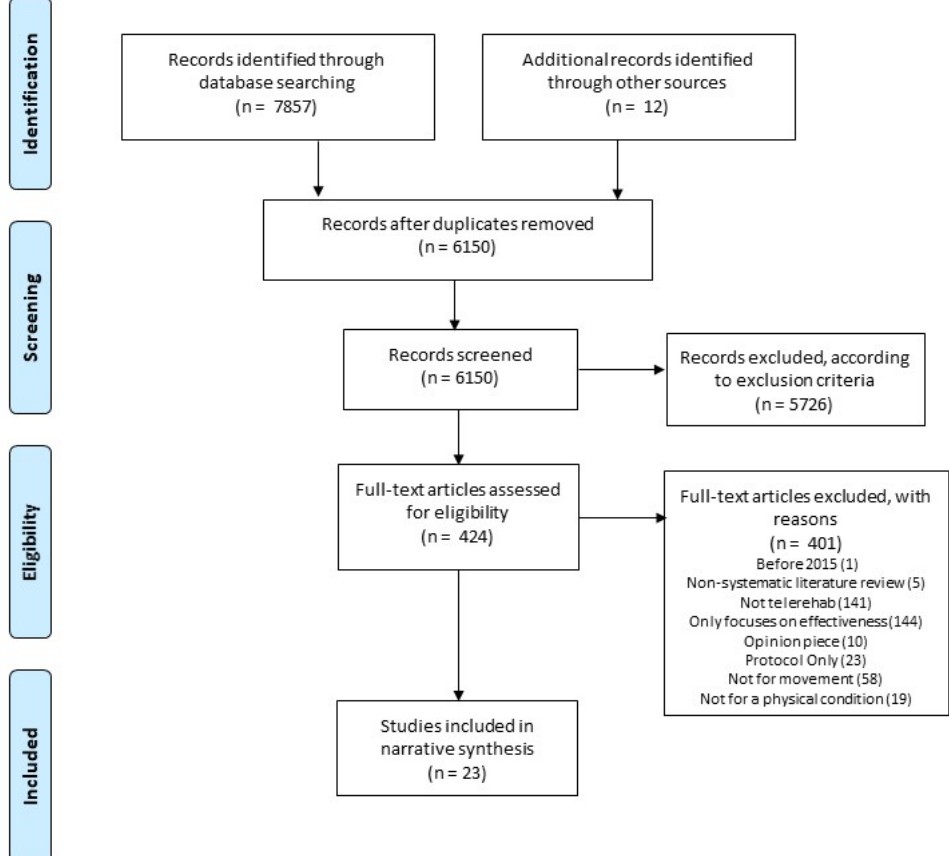

**Figure 1** PRISMA flow diagram that charts the study identification process. PRISMA, Preferred Reporting Items for Systematic Reviews and Meta-Analyses.

## Search strategy

An initial limited search of PubMed and CINAHL was undertaken to estimate the volume of relevant literature and to identify key words to assist in developing search terms. A second search using the developed search terms was undertaken (supported by a review-specialist librarian) and adapted across each included information source (PubMed, CINAHL, PsychInfo, Cochrane, Embase, Web of Science, PEDro, UK Health Forum, WHO, National Archives and NHS England). Due to the rapid nature of this review, study authors were not contacted for further information in cases of ongoing or uncompleted studies. Please see supplementary materials (online supplemental materials 2 and 3) for search strategy terms.

## Study selection

Following the search, all identified references were imported into Endnote[12] (a reference management software). After removal of duplications within EndNote, references were uploaded to the online Rayyan tool[13] (a review organisation tool). Titles and abstracts were screened for assessment against the inclusion criteria for the review. The full text of potentially eligible studies were retrieved and assessed in detail against the inclusion criteria. Full-text studies that did not meet the inclusion criteria were excluded and are displayed in a PRISMA flow chart (see figure 1). Any disagreements that arose

between the reviewers were resolved through discussion. As this review is a rapid scoping review, no quality or risk of bias assessment was undertaken.

## Data extraction

Data extraction of included studies was conducted by the review team. Data were charted using a customised Excel spreadsheet. The data extracted specific information pertaining to the review objectives and key information for each article were charted as per JBI Guidance.[7] This included telerehabilitation contexts, types of telerehabilitation, guidance and training, and any recommendations for implementing telerehabilitation. Any disagreements between reviewers were resolved through discussion.

## Data synthesis

The narrative synthesis of the findings from included studies, policies and guidance documents was structured according to the review objectives. Findings were presented narratively, aided by appropriate tables and figures. Recommendations in the literature were analysed using descriptive qualitative analysis[14] to identify common themes. Only recommendations focusing on clinical implementation or technological improvements were analysed (recommendations for future research within the included studies were not analysed).

Mentions of the 'provider' refer to the individual providing the telerehabilitation, such as health or social care practitioners. Mentions of the 'client' refer to the individual receiving the telerehabilitation, that is, the patient.

## RESULTS

Twenty-three articles were included in the scoping review analysis (figure 1; see table 1 for article characteristics). Eleven[15–25] of the included articles were original research studies, and three were systematic review studies.[26–28] The remaining nine[29–37] were guidance documents produced by health institutions or health experts. Guidance documents are highlighted in grey and systematic review articles are highlighted in yellow in table 1. Eleven articles were specified for adults[15–19 21–24 26 27] (≥18 years), one article was specified for children[20] (≤18 years), and 11 articles did not specify age.[25 28–37]

The 14 included research studies and systematic reviews had the following general aims: one study assessed use and perceptions of telerehabilitation[15]; one study examined costs of implementing telerehabilitation[16]; one study assessed the development of a telerehabilitation technology[17]; three studies systematically reviewed telerehabilitation literature[26–28]; four studies assessed the feasibility of a telerehabilitation technology[18–21]; and four studies examined the effectiveness of the telerehabilitation technology.[22–25]

### Contexts

Client setting was reported in 13 of the 23 articles, where only one article did not include a home setting and instead only used general clinical sites[20] (table 2). The client support environment (ie, who was with the client) was reported in 13 articles (table 2), where most clients were alone (n=5). Only one article specifically reported that carers were present during the consultation or intervention. Table 3 shows the wide range of providers using telerehabilitation in people with physical disability, most being a physiotherapist. Of note, all non-qualified assistants were accompanied by a qualified health professional throughout the telerehabilitation sessions. This article[20] described the development and preliminary usability, from the provider's perspective, of a tablet-based interactive movement tool. Provider setting was reported in six articles, and all were telerehabilitation technologies used by providers in secondary care (eg, an outpatient clinic).[17 19 21–23 25]

### Presence of COVID-19

Only articles published in 2020[16 19 23 29–37] (n=12) were considered for COVID-19 specific information. Of these, two articles refer to COVID-19 related management: guidance documents developed by AHPScot[29] and the Health and Social Care Board Northern Ireland.[30] The COVID-19 related recommendations from these guides are detailed in the Recommendations section. It

is noteworthy that these guidance documents refer to COVID-19 in general and are not specifically related to telerehabilitation use in people with movement impairment, despite these documents generally considering the use of telerehabilitation for this purpose. Teleswallowing Ltd[31], Middleton *et al*[16] and two articles from the Charted Society of Physiotherapy[33 34] also refer to COVID-19 but only state that telerehabilitation is useful to decrease the spread of COVID-19; these articles do not refer to management of patients with COVID-19.

### Types of telerehabilitation

Table 4 shows the aims of the telerehabilitation interventions examined in the primary research studies. Nine guides,[29–37] three systematic reviews[26–28] and two primary studies[18 25] are excluded from table 4 as they did not evaluate a specific telerehabilitation technology. Most (five of the nine studies) telerehabilitation interventions aimed to assist in management of a movement condition; however, some cited additional aims such as optimising adherence,[15] improving quality of life,[24] monitoring and assessing client satisfaction.[23] One intervention provided training in using a wheelchair.[22]

We aimed to summarise the various platforms used for delivery of telerehabilitation, but this proved difficult, as some articles did not have sufficient detail. For example, one article[16] reports a tablet app but does not specify whether video was involved or whether it just used a chat/telephone function. Accepting these limitations, the most common delivery system appeared to be via video (eg, video call, Skype, video-recording, etc), followed by apps (on tablets or mobiles) and telephone calls. Fifteen of the telerehabilitation platforms were synchronous[16 17 19 20 22–24 28–31 33 35–37] (ie, in real-time) and seven platforms used both synchronous and asynchronous[15 18 21 26 27 32 34] (ie, not in real-time) methods. Only one article used only an asynchronous platform where providers assessed physical ability from a video-recording of the client.[25]

Of the 11 primary research studies, five[15 17–19 22] specified neither number of sessions nor session duration, one specified just number and one specified just duration. The number of sessions ranged from 1 to 36 (n=2 for 1 session[20 25]; n=1 for 16 sessions[23]; n=1 for 28 sessions[21]; n=2 for 36 sessions).[16 24] Three studies included 60 min sessions,[20 23 24] one included 50 min sessions[16] and one included 100 min sessions.[21] The variability seen here is reflective of that seen in face-to-face rehabilitation sessions.

### Guidance

Specific guidance on how to use telerehabilitation technology was supplied to providers in 12 articles,[20 22 25 29–37] of which three were studies[20 22 25] and the remaining were guidance documents.[29–37] A guidebook or manual was provided in 10 articles,[20 22 25 29–31 33–35 37] while two articles provided a checklist.[32 36] No guidance was supplied in

**Table 1** Characteristics of the included articles

| Author(s) | Date | Aim | Country origin | Design | Total N | Physical condition |
|---|---|---|---|---|---|---|
| Al Rajeh et al[15] | 2019 | Explore the use of telehealth and assess the perceptions of clinicians employing telehealth and specifically alarm settings on technology. | UK | Cross-sectional | 65 | Chronic obstructive pulmonary disease |
| Amatya et al[26] | 2015 | Investigate the effectiveness and safety of telerehabilitation. | Europe and USA | Systematic review | 564 | Multiple sclerosis |
| Anton et al[17] | 2018 | Develop and examine the Kinext Telerehabilitation System for physical conditions. | Spain and Australia | Feasibility | 18 | Shoulder disorder; hip replacement |
| Block et al[27] | 2016 | Perform a systematic review of studies using remote physical activity monitoring in neurological diseases. | USA | Systematic review | 943 | Neurological diseases |
| Calvaresi et al[18] | 2019 | Examine the feasibility of using artificial intelligence for telerehabilitation. | Italy | Mixed-methods | 10 | General physical conditions |
| GiesbrechtandMiller[22] | 2019 | Evaluate the effect of a mHealth wheelchair skills training programme on clinical outcomes among older adult manual wheelchair clients. | Canada | RCT | 18 | Physical disability needing wheelchair |
| Grau-Pellicer et al[23] | 2020 | Investigate the effectiveness of a mHealth app. | Spain | RCT | 41 | Ischaemic or haemorrhagic stroke |
| Guerra de Oliveira Gondim et al[24] | 2017 | Evaluate the effects of individualised orientation and monitoring by telephone in a self-supervised home therapeutic exercise programme on signs and symptoms of Parkinson's disease and quality of life. | Brazil | RCT | 28 | Parkinson's disease |
| Hasenohrl et al[19] | 2020 | Assess the feasibility and acceptance of orthopaedists prescribing individualised therapeutic exercises via a smartphone app to patients. | Austria | Qualitative | 27 | Back pain |

**Table 1** Continued

| Author(s) | Date | Aim | Country origin | Design | Total N | Physical condition |
|---|---|---|---|---|---|---|
| Levac et al[20] | 2018 | Develop and conduct a usability evaluation of the Fun, Interactive Therapy Board (FITBoard), a movement toy for children with disabilities. | USA | Observational | 7 | General movement conditions |
| Teleswallowing Ltd[31] | 2020 | Provide evidence-based guidance on designing and establishing telerehabilitation services for dysphagia. | UK | Guidance | N/A | Dysphagia |
| Mani et al[28] | 2017 | Systematically explore and summarise the validity and reliability of a telerehabilitation-based physiotherapy assessment. | Malaysia | Systematic review | NS | MSK conditions |
| Middleton et al[16] | 2020 | Describe the process and cost of delivering a synchronous telehealth exercise programme. | USA | Observational | 1 | Stroke, hypertension and diabetes |
| Paneroni et al[21] | 2015 | Test the feasibility, adherence and satisfaction of a home-based reinforcement telerehabilitation programme. | Italy | Observational | 36 | Chronic obstructive pulmonary disease |
| Charted Society of Physiotherapy[33] | 2020 | Provide guidance for physiotherapists on digital tools to support service delivery. | UK | Guidance | N/A | General physical conditions |
| Charted Society of Physiotherapy[32] | 2020 | Provide guidance for physiotherapists on telephone triage of patients with musculoskeletal conditions. | UK | Guidance | N/A | MSK conditions |
| Charted Society of Physiotherapy[34] | 2020 | Provide practical advice for physiotherapists and support workers to implement remote consultations. | UK | Guidance | N/A | General physical conditions |
| Royal College of Speech and Language Therapists[36] | 2020 | Provide brief tips for telerehabilitation implementation revolved around security. | UK | Guidance | N/A | General physical conditions |
| East Lancashire Hospitals NHS Trust[37] | 2020 | Provide brief tips for telerehabilitation implementation for dysphagia. | UK | Guidance | N/A | Dysphagia |

Continued

**Table 1**  Continued

| Author(s) | Date | Aim | Country origin | Design | Total N | Physical condition |
|---|---|---|---|---|---|---|
| Venkataraman et al[25] | 2016 | Compare Berg Balance Scale ratings using videos with differing transmission characteristics with direct in-person rating. | USA | Psychometrics study | 45 | General motor control condition |
| NI Health and Social Care Board NI[30] | 2020 | Amalgamate evidence and guidance on the conduct of virtual consultations so that it is easily accessible for AHPs. | UK | Guidance | N/A | General physical conditions |
| AHPScot[29] | 2020 | Guide AHPs in NHS Scotland during remote consultations (telephone or video) with MSK patients during the current COVID-19 pandemic. | UK | Guidance | N/A | MSK conditions |
| Royal College of Occupational Therapists[35] | 2020 | Guide occupational therapists regarding implementation of virtual assessments and reviews. | UK | Guidance | N/A | General physical conditions |

Guidance documents are highlighted in grey and systematic review articles are highlighted in yellow.

AHPs, allied health professionals; MSK, musculoskeletal; N/A, not applicable; NHS, National Health Service; NI, Northern Ireland; NS, not specified; RCT, randomised controlled trial.

**Table 2** Cross-tabulation of client support and client setting information extracted from included articles

| | | Article type | Client support environment | | | |
|---|---|---|---|---|---|---|
| | | | Not specified | Client with group | Client alone | Client with carer |
| Client setting | Not specified | Primary Research | 2[15 17] | – | 1[25] | – |
| | | Reviews | – | 1[28]* | 1[28]* | – |
| | | Guidance Documents | 6[31 32 34–37] | – | – | – |
| | At home | Primary Research | – | – | 4[16 19 21 22] | 1[24] |
| | | Reviews | – | – | – | – |
| | | Guidance Documents | – | – | – | – |
| | Home and community | Primary Research | 1[18] | – | – | – |
| | | Reviews | – | 2[26 27]* | 2[26 27]* | – |
| | | Guidance Documents | 1[33] | 2[29 30]* | 2[29 30]* | – |
| | General clinic sites | Primary Research | 1[20] | – | – | – |
| | | Reviews | – | – | – | – |
| | | Guidance Documents | – | – | – | – |
| | Rehabilitation unit and at home | Primary Research | – | 1[23] | – | – |
| | | Reviews | – | – | – | – |
| | | Guidance Documents | – | – | – | – |

*Five articles (three reviews[26–28] and two guidance documents[29 30]) had participants complete the telerehabilitation alone and in a group; these articles have been added to both the 'Client was alone' and 'Client with group' categories.

seven articles,[15 16 19 21 23 24] and five studies did not mention whether guidance was provided or not.[17 18 26–28]

### Training

Guidance documents and systematic reviews were not included in training analysis as these articles had no participants to whom to supply training. Specific training on how to use telerehabilitation technology was supplied to providers in 4 out of 11 primary research studies.[16 20 22 25] The training was provided in-person in three of these studies,[16 20 22] while the remaining study provided training via a video tutorial.[25] No training was supplied in five studies,[15 19 21 23 24] and two studies did not mention training.[17 18]

### Recommendations

Six themes and 11 subthemes were identified within the recommendations: administrative advice, challenges, governance, social support, technology improvements and COVID-19 (see table 5). Recommendations were also analysed to extract procedural guidance that providers can perform in preparation for, during and after a

**Table 3** List of telerehabilitation administrators extracted from included articles

| Telerehabilitation administrator* | N† | N‡ | N§ |
|---|---|---|---|
| Physiotherapist | 9[15–18 20 21 23–25] | 1[28] | 1[34] |
| Occupational therapist | 3[20 22 25] | – | 1[35] |
| Medic | 3[15 19 21] | 1[27] | – |
| Nurse | 2[15 21] | – | – |
| Speech and language therapist | – | – | 2[31 37] |
| General clinicians | – | – | 2[32 33] |
| AHPs | – | – | 2[29 30] |
| Not specified | 1[36] | 1[26] | – |
| Assistants | 2[16 25] | – | – |
| Physiologist | 1[15] | – | – |

*Telerehabilitation administrators overlap in articles; therefore, total N does not sum to total included articles.
†N for primary research studies.
‡N for systematic review studies.
§N for guidance documents.

**Table 4** List of telerehabilitation aims and platforms reported in the included articles

| Telerehabilitation aim | N | References |
|---|---|---|
| MCM | 5 | 16 17 19–21 |
| MCM and adherence | 1 | 15 |
| MCM and quality of life | 1 | 24 |
| MCM and client satisfaction | 1 | 23 |
| Training for assistive device use | 1 | 22 |

MCM, movement condition management.

telerehabilitation session. This guidance is summarised in supplementary materials (online supplemental material 4) as it extended the review beyond its key message.

## Administrative recommendations

Three articles[30 32 36] provided advice on administrative duties that providers should perform when conducting telerehabilitation. The advice revolved around four subthemes:

1. General advice involved general administrative duties such as maintaining record-keeping standards and identifying the specific resources a provider will need when engaging in telerehabilitation (eg, number of

**Table 5** Summary of themes identified during qualitative analysis of article recommendations

| Analysis themes | Description |
|---|---|
| Administrative recommendations<br>  General<br>  Client specific<br>  Technology specific<br>  Data security | Advice revolved around administrative duties, such as record keeping, technology maintenance or data security. |
| Challenges<br>  Care provision<br>  Social factors<br>  Technology issues | Challenges that providers may encounter when conducting telerehabilitation services. |
| Governance | Ethical advice, such as client consent and/or assessing confidentiality. |
| Support mechanisms<br>  Providers<br>  Carers<br>  Provider assistants<br>  Non-clinical staff | Advice that emphasised the important role of support from various sources involved in providing telerehabilitation care to the client. |
| Technology improvements | Suggestions that may improve the use and function of a telerehabilitation technology. |
| COVID-19 | Actions providers are advised to conduct to address COVID-19 risks. |

sessions needed, appropriate length of session considering technology set-up, clinical outcome measures needed, etc).

2. Client-specific advice involved duties that required interactions with the client, such as documenting all client conversations and noting care that would have been provided if the intervention was delivered face to face but was not done so due to the restrictions of telerehabilitation.

3. Technology-specific advice involved ensuring technology was up to date and that the latest version of any software was being used.

4. Data security emphasised that providers must minimise identifiable information that is shared digitally, ensure that all actions comply with information governance and only use technology recognised by the institute through which they provide their care services.

## Challenges

Six articles[15 19 23 25 28 30] highlighted challenges when conducting telerehabilitation, which involved three subthemes:

1. Care provision considers the resources needed for telerehabilitation, highlighting that some clinical assessments may be too complicated to conduct effectively at a distance and that, if this complication is not recognised by inexperienced telerehabilitation providers, the quality of care provided may be reduced.

2. Social factors identified social challenges, where unsatisfactory video etiquette, minimal rapport building and socioeconomic disadvantages have the potential to reduce the quality of care provided via telerehabilitation.

3. Technology issues relate to technology failures or interruptions, such as inadequate internet connection, low camera/audio quality and false alarms (on wearable devices) that result in a reduced quality of care.

## Governance

Six articles[29 30 34–37] provided recommendations relating to governance issues. The guidance highlighted that all clients must be informed about the following: telerehabilitation is voluntary; potential risks when engaging in telerehabilitation; what will happen with their data; and what reasonable actions they should take to ensure the security of their technology. This theme also emphasised the importance of consent, where informed consent must be taken explicitly from clients when video/audio recording or taking screen shots. If required, clients must be asked whether they consent for their trusted others (eg, carer or family) to attend the session with them. However, one article[30] stated that consent was implied when clients enter the telerehabilitation session.

## Support mechanisms

Six articles[21 23 24 28 31 37] recommended support mechanisms to be set in place for clients when providing telerehabilitation. This support involved four subthemes:

1. Providers to offer support beyond the main goals of telerehabilitation, such as reassurance to clients who are anxious using technology for their rehabilitation.
2. Carers to offer clients in-person support during delivery of telerehabilitation sessions. For instance, provision of physical assistance to safely enable the client to achieve correct posture or movement or to assist the client with using the technology, such as positioning of the camera.
3. Provider assistants to assist qualified staff by visiting clients in person to assist telerehabilitation, such as undertaking outcome measures face to face, when required and safe to do so.
4. Non-clinical staff, such as administrators and senior management, to provide essential support in successful coordination and implementation of telerehabilitation, emphasising the importance of effective communication between support staff and providers.

### Technology improvements

Eight articles[15] [17–20] [26–28] recommended technology improvements for telerehabilitation purposes. For instance, communications should be synchronous, and technology must have a client-friendly interface that satisfies both the client and the provider. Wearable technologies relevant to posture and movement (eg, to measure steps or heart rate) must be precise. Technology would especially benefit from improved monitoring capabilities, such as real-time movement tracking and activity adherence monitoring. If technology provided smart data (eg, real-time measures of muscle strength), it should be accompanied by useful and actionable information. Reminders to engage in self-care activities would be useful for clients during their rehabilitation period. A built-in anatomical map to assist clients in showing the location of pain, tenderness or other sensations was also suggested.

### COVID-19

Two articles (both guidance documents)[29] [30] made recommendations to address COVID-19 concerns. When establishing whether the client needs telerehabilitation or face-to-face care, professionals were advised to identify the client's COVID-19 status. Suggested identifications included: tested positive and self-isolating, has symptoms and self-isolating, has come into contact with someone who tested positive and is now self-isolating, shielding or no current cautions outside of government restrictions. Of note, this is an area of rapid expansion in the literature.

### DISCUSSION

The 23 articles included in this rapid scoping review revealed two main gaps in the literature: (1) telerehabilitation guidance (from research studies and guidance documents) is not specific to movement-related rehabilitation despite articles stating this explicitly as their aim and (2) neither guidance nor training were given to

providers in most research studies evaluating a telerehabilitation technology.

### The lack of specific guidance

The vast majority of the existing recommendations relate predominantly to communication or governance elements of telerehabilitation and lack specific guidance about issues pertinent to movement impairment. Several elements that are key to effectively implementing telerehabilitation for people with physical disability are missing. When assessing movement remotely, one would expect guidance on ensuring patient safety, triaging patients and specific steps to decide whether a patient needs face-to-face care. More importantly, the identified guidance does not effectively address the limitations of undertaking physical assessments remotely, for example, if the patient is unable to move their body themselves if their movement is inadequate for assessment. Remote physical examinations are complex and need specific guidance on implementation; this is reflected in recent research that highlights physical examinations as one of the main challenges for remote assessments.[38 39] There was also minimal information on telerehabilitation related to COVID-19. This is unsurprising given the novelty of this infection. Overall, current telerehabilitation guidance for physical disabilities focuses on general telehealth guidance rather than being specific to movement impairments. Although issues such as data security, video etiquette and organisation are important, they are not enough to assist providers in effective telerehabilitation implementation for people with movement impairments.

### The minimal focus on providers

Of the 11 included primary studies and three systematic reviews, only three[20 22 25] provided guidance to providers. Similarly, only four research studies[16 20 22 25] described the provision of training to providers. Neglecting providers' skills in using technology effectively (ie, digital literacy) potentially undermines the usefulness of any telerehabilitation technology that was evaluated.[40 41] Furthermore, studies mainly report the clients' and not the providers' context of using telerehabilitation (eg, whether the provider was from primary, secondary or tertiary care). The generalisability of the telerehabilitation technology examined in the study is thus unclear, as certain technologies may only be effective in particular contexts.[42] Future studies should carefully consider providers' digital literacy and context to better understand the effectiveness of telerehabilitation technology. Digital literacy and context are not the only important factors for telerehabilitation implementation. Healthcare providers need appropriate organisational infrastructures (eg, effective IT support) and a sufficient workflow integration to effectively implement technology.[43] They also reported that an easy-to-use interface is important, which aligns with the review's finding of needing a friendly client interface for technological improvements.

## Digital equity

Although digital equity was not part of this review's objectives, it is important to address. The included articles provide information based on the assumption that all clients (and providers) have both access and skills to engage in technology for remote assessments. Many of those currently digitally excluded are likely to become increasingly neglected from remote health services, thereby exacerbating existing health inequalities.[44] Future studies should acknowledge equity of access when examining telerehabilitation technologies. Future guidance documents should provide information for when clients do not have access to (or decline to use) telerehabilitation.

## Comparisons with other studies

To our knowledge, this is the first scoping review of telerehabilitation guidance and training for assessing physical disabilities within daily clinical practice. Previous reviews mostly examine effectiveness within the context of research studies and conclude that further provider and client training is needed to adequately implement telerehabilitation for physical assessments.[45 46] These studies conclude that telerehabiltiation technology has potential to deliver effective physical disability care; the current state of the art is insufficient for accurate remote movement assessment.[47] Our review confirms the need for further provider training and additionally emphasises the lack of specific movement-related guidance.

## Study limitations

This scoping review has several limitations. First, many telerehabilitation guidance documents for physical disabilities are likely unpublished. Although this review conducted an extensive search within grey literature sources, guidance documents and other relevant studies have likely been missed. Second, the search strategy was biased towards the English language. Despite no language restrictions during the search and the use of language translation software, relevant articles have likely been missed. This review also included a date restriction, where no articles prior to 2015 were included in the literature search in order to exclude outdated technological information. Yet, it should be noted that this may have also excluded some relevant aspects on guidance and training. Finally, this review was restricted by its rapid, cross-sectional approach. The authors acknowledge that some articles within this review may have been updated since the initial search and others have likely emerged since. This review may serve as a foundation for future research and development of guidance documents to provide information that addresses the gaps identified above.

## CONCLUSION

There is a clear need for specific telerehabilitation guidance for physical disabilities, particularly for effective practice of remote physical assessments. Studies examining telerehabilitation technologies should deliver guidance and training to providers as well as document the context of providers if those technologies are to be effectively implemented. The catalyst that is the COVID-19 pandemic has forced the uptake of remote rehabilitation services, which are likely to persist beyond the pandemic. The development and maintenance of efficient telerehabilitation will require detailed guidance and active performance monitoring for ongoing improvement of existing guidance, without which remote physical assessments may result in suboptimal management.

**Acknowledgements** We would like to sincerely thank Chris Johns from the University of Plymouth, who provided library support during the initial phase of this scoping review, and particularly assisted in developing the search strategy. Thanks to Sarah Chatfield, who undertook the initial limited search of the literature.

**Contributors** KA planned and conducted the literature search, organised the review team, analysed and reported the results and produced the initial manuscript. JAF conceptualised the study idea and supervised study processes. All authors reviewed the articles from the literature search, extracted data, contributed to data interpretation and critically evaluated the manuscript. The corresponding author, KA, attests that all listed authors meet authorship criteria and that no others meeting the criteria have been omitted.

**Funding** This work was supported by UKRI-NIHR (MRC Section), Covid-19; Reference MR/V021060/1.

**Disclaimer** The funder had no role in study design, data collection, data analysis or writing of the report.

**Competing interests** None declared.

**Patient consent for publication** Not required.

**Ethics approval** This study is a scoping literature review and did not involve participant data collection. Therefore, ethical approval was not required.

**Provenance and peer review** Not commissioned; externally peer reviewed.

**Data availability statement** No data are available.

**ORCID iDs**
Krithika Anil http://orcid.org/0000-0002-8027-1665
Ray B Jones http://orcid.org/0000-0002-2963-3421

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
