## [Reviewer comments · BMJ Open]

ARTICLE DETAILS

TITLE (PROVISIONAL)	The scope, context, and quality of telerehabilitation guidelines for physical disabilities: a scoping review
AUTHORS	Anil, Krithika; Freeman, Jennifer; Buckingham, Sarah; Demain, Sara; Gunn, H; Jones, Ray; Logan, Angela; Marsden, Jonathan; Playford, Diane; Sein, Kim; Kent, Bridie

VERSION 1 – REVIEW

REVIEWER	Krasovsky, Tal University of Haifa
REVIEW RETURNED	17-Mar-2021

GENERAL COMMENTS	This manuscript presents a rapid scoping review on telerehabilitation guidelines for movement assessment among people with physical disabilities, including people recovering from COVID-19. While the topic is timely and important, and the work performed is indeed extensive, several points should be considered in order for this manuscript to be published. First, the definition provided for telerehabilitation is missing a key component, that of distance. For example, the US Department of Health and Human Services defines telehealth as “the use of electronic information and telecommunication technologies to support long-distance clinical health care, patient and professional health-related education, public health and health administration”. The long-distance provision of care component is missing from the definition provided, a point which may affect study selection. For example, the paper by Levac et al. does not present a telerehabilitation solution, rather a technological solution for movement assessment. An additional point which should be considered is the treatment of the COVID-19 pandemic. Importantly, telerehabilitation is gaining momentum not only due to rehabilitation of people with/post COVID, but also due to the need for physical distancing in other populations. In various populations of people with physical disabilities, telerehabilitation under extreme physical isolation may not be similar in its characteristics (e.g. aims) to telerehabilitation performed under different, less stressful conditions. In my opinion, this distinction should be emphasized in the paper. Furthermore, with the search ending in August 2020, rather early in the course of the pandemic, it is not surprising that very little work was published regarding COVID-19 and telerehabilitation. I assume that furthering the search by several months may change this conclusion. Alternatively, less focus should be given to the pandemic in this context. The selection of 2015 as the starting date for the search needs to be explained, especially since the reason provided is that technology older than 2015 is considered out-of-date. However, as the authors themselves find, the technology which governs
--

	telerehabilitation applications is still video. Thus, older papers may be still relevant and potentially important. A minor point has to do with the exclusion criteria: the authors state that papers which focus only on effectiveness have been excluded (N=144, PRISMA flow diagram), but do mention several papers which evaluate effectiveness in table 1. It is not clear why would articles which focus on effectiveness be excluded at all? Some papers may have been overlooked in the search – e.g. Cottrell et al. 2017.
--	--

REVIEWER	Hailey, David University of Alberta, Public Health Sciences
REVIEW RETURNED	06-Apr-2021

GENERAL COMMENTS	Abstract. The brief Introduction includes a useful illustration of questions related to introduction of telerehabilitation. Minor edit needed Page 4 The first paragraph in Methods might be better placed in the Introduction P5 Publications prior to 2015 were not included because of the recent rapid development of technologies. Possibly this might be a limitation in that not all recent technologies will be in widespread use and relevant guidance provided in earlier publications might well be relevant to current and future practice. Lines 50-51 perhaps the explanation of the Rayyan tool is not needed P 10 Suggest moving the sentence on provider setting to the end of the paragraph on Contexts I found it difficult to match the words on client support environment to the details in Table 2. It may be helpful to include information from the table footnote in the text. There appear to have been 6 studies where the clients were alone not 5. P 12 the paragraph on COVID-19 provides a useful clarification. P 16 Details are provided in the summary on lack of specific guidance in the reviewed studies on application of telerehabilitation to movement impairment. This situation gives significant limitations to the usefulness of this approach. Such challenges are not new to telerehabilitation and relevant advice on some aspects may be found in earlier literature. Lines 35-36 Inclusion of a reference supporting this point on provider skills would be helpful. 44 The need for organisational infrastructure to support health care providers is well established. P 17 The need for telerehabilitation guidance, noted in the Conclusion, is well supported by the material presented in the manuscript. What will need to be addressed, as in other areas of telehealth, is the reality of applying guidelines to routine provision of services. Continuity of effective telerehabilitation will require appropriate quality control and active monitoring of performance by both service providers and their patients.
--

VERSION 1 – AUTHOR RESPONSE

Reviewer 1 Comments

This manuscript presents a rapid scoping review on telerehabilitation guidelines for movement assessment among people with physical disabilities, including people recovering from COVID-19. While the topic is timely and important, and the work performed is indeed extensive, several points should be considered in order for this manuscript to be published.

Response: We are pleased the reviewer finds our paper important, and we thank the reviewer for the comments that we have addressed below.

First, the definition provided for telerehabilitation is missing a key component, that of distance. For example, the US Department of Health and Human Services defines telehealth as “the use of electronic information and telecommunication technologies to support long-distance clinical health care, patient and professional health-related education, public health and health administration”. The long-distance provision of care component is missing from the definition provided, a point which may affect study selection. For example, the paper by Levac et al. does not present a telerehabilitation solution, rather a technological solution for movement assessment.

Response: We thank the reviewer for their comment. We did not include long-distance within the definition for telerehabilitation as we consider telerehabilitation to be a subset of the more generic telemedicine, which is simply defined as “the remote diagnosis and treatment of patients by means of telecommunications technology”. Distance is not included in that definition. Furthermore, we used Brennan [2] for our definition of telerehabilitation. They also do not include distance in the definition. While, we of course agree that saving patients from travel is a major benefit of telerehabilitation, that benefit can be achieved for short as well as long journeys.

However, we would not need to change our search strategy if we changed our definition to include distance. This is because “concept 2” of our search term logic grid (see supplementary material “SM1”) uses terms such as “virtual”, “tele*”, “online”, etc. that consequentially implies technology used at a distance.

An additional point which should be considered is the treatment of the COVID-19 pandemic. Importantly, telerehabilitation is gaining momentum not only due to rehabilitation of people with/post COVID, but also due to the need for physical distancing in other populations. In various populations of people with physical disabilities, telerehabilitation under extreme physical isolation may not be similar in its characteristics (e.g. aims) to telerehabilitation performed under different, less stressful conditions. In my opinion, this distinction should be emphasized in the paper.

Response: We have emphasised this point at the end of the first introduction paragraph: “It is especially important to note that the practical application of telerehabilitation will not be the same across all conditions. More complicated conditions (e.g. those with co-morbidity) will likely require additional support than less complicated conditions. This additional consideration further demonstrates the need for comprehensive training and guidance for telerehabilitation. ”.

Furthermore, with the search ending in August 2020, rather early in the course of the pandemic, it is not surprising that very little work was published regarding COVID-19 and telerehabilitation. I assume that furthering the search by several months may change this conclusion. Alternatively, less focus should be given to the pandemic in this context.

Response: We agree with the reviewer that August 2020 was early in the pandemic. This is why we conducted a broad search of telerehabilitation, not just within the context of COVID-19. Although the paper was prompted by COVID-19, we consider telerehabilitation in all circumstances and for whatever benefit. As suggested by the reviewer, we have toned down the emphasis on COVID in the introduction.

The selection of 2015 as the starting date for the search needs to be explained, especially since the reason provided is that technology older than 2015 is considered out-of-date. However, as the authors themselves find, the technology which governs telerehabilitation applications is still video. Thus, older papers may be still relevant and potentially important.

Response: A date limiter was implemented because video call technology (both software and quality of transmission) has improved considerably in the last decade. For example, Facetime was enabled over mobile networks in the UK in 2012: <https://www.wired.co.uk/article/facetime-3g-ios6>

And WhatsApp video calls were introduced in 2016
<https://www.forbes.com/sites/amitchowdhry/2016/11/15/whatsapp-video-calling-launches/?sh=708a45a5e456>

Papers published in 2015 are likely to be referring to technologies from 2014 or before, and therefore 2015 was a reasonable compromise cut-off date for this review. The difference in technology between articles pre and post 2015 would be significant, making the technology guidance/training in pre 2015 articles outdated. Additionally, the use of video calls – as a result of technological advances – started to increase more rapidly from about 2014. (See e.g. figure below of frequency of the word “video call” on NGRAM viewer – gradient increases from about 2014).

A minor point has to do with the exclusion criteria: the authors state that papers which focus only on effectiveness have been excluded (N=144, PRISMA flow diagram), but do mention several papers which evaluate effectiveness in table 1. It is not clear why would articles which focus on effectiveness be excluded at all? Some papers may have been overlooked in the search – e.g. Cottrell et al. 2017.

Response: Our review was interested in the training and guidance regarding telerehabilitation, not whether telerehabilitation was effective. This is because there are instances where telerehabilitation is not a choice, which has been exacerbated by the COVID-19 pandemic. Therefore, this review excluded papers that only focused on effectiveness without information regarding training and guidance. Some papers within Table 1 had effectiveness as their main aim. However, these papers also included training/guidance information relevant to our review aim.

Reviewer 2 Comments

The brief Introduction includes a useful illustration of questions related to introduction of telerehabilitation. Minor edit needed

Response: We thank the reviewer for their comment. We have made minor edits to the introduction for clarity.

Page 4 The first paragraph in Methods might be better placed in the Introduction

Response: We have moved the relevant paragraph to the introduction, just before the section “Review question and objectives”.

P5 Publications prior to 2015 were not included because of the recent rapid development of technologies. Possibly this might be a limitation in that not all recent technologies will be in widespread use and relevant guidance provided in earlier publications might well be relevant to current and future practice.

Response: Reiterating our response to reviewer 1, a date limiter was implemented because video call technology has improved considerably in the last decade. Papers published in 2015 are likely to be referring to technologies from 2014 or before, and 2015 was a reasonable cut-off date. The

difference in technology between articles pre and post 2015 would be significant, making the technology guidance/training in pre 2015 articles outdated.

Lines 50-51 perhaps the explanation of the Rayyan tool is not needed

Response: We have condensed this explanation to "(a review organisation tool)".

P 10 Suggest moving the sentence on provider setting to the end of the paragraph on Contexts

I found it difficult to match the words on client support environment to the details in Table 2. It may be helpful to include information from the table footnote in the text. There appear to have been 6 studies where the clients were alone not 5.

Response: We have moved the provider setting information to the end of the "Contexts" paragraph. We thank the reviewer for identifying the mistake regarding Table 2. The footnote now states that 5 (not 4) articles included clients that were alone and in a group. Thus, clients were alone in 5 studies and not 6. We hope this clarifies Table 2 regarding client support environment.

P 16 Details are provided in the summary on lack of specific guidance in the reviewed studies on application of telerehabilitation to movement impairment. This situation gives significant limitations to the usefulness of this approach. Such challenges are not new to telerehabilitation and relevant advice on some aspects may be found in earlier literature.

Response: We agree that these challenges are not new to telerehabilitation. However, we reason that these challenges may have been better addressed if earlier literature had provided relevant and specific advice and guidance.

Lines 35-36 Inclusion of a reference supporting this point on provider skills would be helpful.

Response: We have added the following references: Kuek et al [41] and Poncette et al [42].

44 The need for organisational infrastructure to support health care providers is well established.

Response: We have changed this sentence to "Healthcare providers need appropriate organisational infrastructures (e.g. effective IT support) and a sufficient workflow integration to effectively implement technology⁴²".

P 17 The need for telerehabilitation guidance, noted in the Conclusion, is well supported by the material presented in the manuscript. What will need to be addressed, as in other areas of telehealth, is the reality of applying guidelines to routine provision of services. Continuity of effective telerehabilitation will require appropriate quality control and active monitoring of performance by both service providers and their patients.

Response: We thank the reviewer for this point and have included it in the "Conclusion": "The development and maintenance of efficient telerehabilitation will not only require detailed guidance, but also active performance monitoring for on-going improvement of existing guidance, without which remote physical assessments may result in sub-optimal management".

VERSION 2 – REVIEW

REVIEWER	Krasovsky, Tal University of Haifa
REVIEW RETURNED	28-May-2021

GENERAL COMMENTS	I am happy with the author responses to the review.
---

REVIEWER	Hailey, David University of Alberta, Public Health Sciences
REVIEW RETURNED	02-Jun-2021

GENERAL COMMENTS	The suggestions on editing and placement of material have been followed and the revisions made are satisfactory. Also additional references have been added. On the issue of not including pre-2015 articles the authors indicate "the difference in technology between articles pre and post 2015 would be significant, making the technology guidance/training in pre 2015 articles outdated " This point is appreciated but some relevant aspects on guidance may have been covered in the earlier literature. Also in response to a comment on p16 that some challenges are not new to telerehabilitation and relevant advice on some aspects may be found in earlier literature it is suggested that "these challenges may have been better addressed if earlier literature had provided relevant and specific advice and guidance." No support is provided for that statement which takes no account of relevant, good quality publications. Consideration could be given to inclusion of a brief comment indicating this as a limitation of the study.
--

VERSION 2 – AUTHOR RESPONSE

We would like to thank the reviewers for their comments on our revised manuscript. Our responses are below, and we look forward to hearing from you in due time.

Reviewer 1 Comments

I am happy with the author responses to the review.

Response: We are pleased the reviewer finds our paper suitable for publication.

Reviewer 2 Comments

The suggestions on editing and placement of material have been followed and the revisions made are satisfactory. Also additional references have been added.

Response: We are pleased the reviewer finds our previous responses satisfactory.

On the issue of not including pre-2015 articles the authors indicate "the difference in technology between articles pre and post 2015 would be significant, making the technology guidance/training in pre 2015 articles outdated " This point is appreciated but some relevant aspects on guidance may have been covered in the earlier literature.

Response: We have addressed this comment, and the one below, by adding the following to the limitations: "This review also included a date restriction, where no articles prior to 2015 were included in the literature search in order to exclude outdated technological information. Yet, it should be noted that this may have also excluded some relevant aspects on guidance and training."

Also in response to a comment on p16 that some challenges are not new to telerehabilitation and relevant advice on some aspects may be found in earlier literature it is suggested that "these challenges may have been better addressed if earlier literature had provided relevant and specific advice and guidance." No support is provided for that statement which takes no account of relevant,

good quality publications. Consideration could be given to inclusion of a brief comment indicating this as a limitation of the study.

Response: As recommended, we have added this to the limitations section (see above response).